# Distribution and characteristics of androgen receptor (AR) in breast cancer among women in Addis Ababa, Ethiopia: A cross sectional study

Endale Hadgu[1,¤a]*, Daniel Seifu[1,¤b], Wondemagegnhu Tigneh[2], Yonas Bokretsion[3], Abebe Bekele[4,¤b], Markos Abebe[5], Thomas Sollie[6], Christina Karlsson[7‡], Mats G. Karlsson[6‡]

1 Department of Biochemistry, School of Medicine, Addis Ababa University, Addis Ababa, Ethiopia,
2 Department of Oncology, School of Medicine, Addis Ababa University, Addis Ababa, Ethiopia,
3 Department of Pathology, School of Medicine, Addis Ababa University, Addis Ababa, Ethiopia,
4 Department of Surgery, School of Medicine, Addis Ababa University, Addis Ababa, Ethiopia, 5 Armauer Hansen research Institute (AHRI), Addis Ababa, Ethiopia, 6 School of Medical Sciences, Orebro University, Orebro, Sweden, 7 School of Health Sciences, Orebro University, Orebro, Sweden

☯ These authors contributed equally to this work.
¤a Current address: Department of Biochemistry, St. Paul's Hospital Millennium Medical College, Addis Ababa, Ethiopia
¤b Current address: University of Global Health Equity, Kigali, Rwanda
‡ These authors also contributed equally to this work.
* endale.hadgu@sphmmc.edu.et

**Data Availability Statement:** All relevant data are within the manuscript and its Supporting Information files.

## Abstract

Evaluation of the role of androgen receptor (AR) in the biology of breast cancer is an emerging area of research. There are compelling evidences that AR expression may be used to further refine breast cancer molecular subtyping with prognostic and therapeutic implications. Many studies indicated co-expression of AR with the hormonal receptors in breast cancer has a favorable prognosis. AR is also investigated by many researchers as a potential therapeutic target in treatment of breast cancer. Studies on the frequency and distribution of AR in breast cancer among Africans is barely available. Given the heightened interest to understand its role in breast cancer, we determined AR expression and assessed its association with clinico-pathological parameters among Ethiopian women. In this study, 112 newly diagnosed patient with invasive breast cancer at Tikur Anbessa Specialized Hospital were enrolled. Immunohistochemical assessment of AR, ER, PR, Ki67 and HER2 were performed using tissue microarrays (TMA) constructed from their primary tumor block. Out of the 112 participants, 91 (81%) were positive for AR expression and the remaining 21 participants (19%) were negative for AR expression. Expression of AR in ER+, HER2+ and TNBC cases were 93%, 83% and 48% respectively. Our study reveals AR is expressed in a significant number of breast cancers patients and this may indicate that breast cancers cases in Ethiopia have favorable prognosis and could benefit from progresses in AR targeted treatments. Since AR expression has important consequences on the prognosis and treatment of breast cancer, further studies with an increased number of participants is necessary to confirm our reports.

**Funding:** This work is funded by Addis Ababa University, School of Graduate Studies, thematic research group "clinico-epidemiological characterization of breast cancer in Ethiopia" and Armauer Hansen Research Institute (AHRI). AHRI supported the study through funding obtained from Swedish International Developmental Agency (SIDA), Sweden.

**Competing interests:** The authors declare that they have no competing interests.

## Introduction

Breast cancer is the most common malignancy and is the major cause of cancer death among women worldwide [1]. It is a heterogeneous disease with a variety of subtypes, each characterized by distinct clinical, pathologic and molecular features [2]. The widely accepted predictive or/and prognostic factors in breast cancer include steroid or growth hormone receptors including estrogen receptor (ER), progesterone receptor (PR) and human epidermal growth factor receptor type 2 (HER-2) [3, 4]. Understanding the role of steroid hormone receptors in breast cancer has led to the development of hormonal therapy which generally have less side effect when compared to chemotherapy [3, 5]. In addition, targeted therapies to human epidermal growth factor receptor type 2 (HER-2) positive tumors were developed through continuous improvement in the understanding of the molecular biology of breast cancer [6].

The role of androgen receptor (AR) in prostate cancer is well established and AR targeted drugs are currently part of the standard care positively affecting the course and outcome of the disease [7, 8]. However, the role of AR in breast cancer is emerging only recently because of the increased interest among researchers in breast cancer to understand a disease which is so heterogenous in its molecular feature and limited treatment options [7–9]. An increasing number studies evaluated AR as a useful marker for the further refinement of breast cancer molecular subtype and as an emerging clinical target [10, 11]. Some authors suggested assessment of Androgen receptor in breast cancer in the routine diagnosis, as part of a quadruple panel alongside the assessment of ER, PR, and HER2 to serve as an additional predictive and/or prognostic marker [11, 12].

AR is expressed in all stages of breast cancer (in-situ, primary and metastatic disease) [13] and several studies show AR may play different prognostic role in ER-positive and ER-negative breast cancers [7, 8, 14]. In ER-positive breast cancer, AR was reported to predict favorable disease outcome consistently. Co-expressing AR and ER in breast cancer improved disease-free survival (DFS) and overall survival (OS) significantly [8, 14, 15]. In ER-positive tumors, AR seems to inhibit the cellular proliferation induced by estradiol and to have a favorable prognostic value [8, 12, 15, 16]. However, in ER-negative breast cancers AR may be able to drive disease progression and may be linked to poor prognosis [10, 17, 18].

Preclinical and clinical studies conducted in recent years are supporting the role of AR-targeting treatment in the management of breast cancer [9, 13, 16, 19]. Therefore, both AR antagonists and AR agonists will likely become useful and safe options of treatment in various breast cancer subtypes particularly in combination with other agents already proved to be beneficial in treating breast cancer, but only the ongoing and future prospective clinical trials will allow us to establish which agents are the best options in every specific condition [8, 9, 13, 16, 20–22].

Most of the studies conducted to characterize the expression of AR so far are done in the western countries and shows AR is expressed in approximately 80 and 60% of primary and metastatic breast tumors, respectively. The frequency of AR expression in those studies varies across the clinical subtypes, approximately 84–95% in ER+ tumors, 50–63% in ER−/HER2 + tumors, and 10–53% in TNBC [8, 14, 17, 21–23].

Only few studies have been conducted in Africa to characterize Androgen receptor (AR) in breast tumor. One of the study done among Tanzanians by Bravaccini et.al in 2018 [24] reported an overall frequency of AR-positivity at >1% cut-off value to be 66% and another done among Ghanaians by Proctor E et.al. in 2015 [25] reported an overall frequency of AR-positivity at >10% cut-off value to be 44%. This study is a continuation of our previous work in molecular classification of breast cancer among Ethiopian women [26]. We observed that breast cancer in Ethiopian patients at time of diagnosis are majorly hormone receptor positive unlike other African patients and we observed that the biological characteristics of breast

cancer among Africans is not homogenous which has a huge impact on breast cancer prevention and treatment in the continent. Therefore, our present study aimed to determine the prevalence of AR expression and its association with clinicopathological parameters in Ethiopian breast cancer patients.

## Materials and methods

The study was approved by Institutional Review Board (IRB) of Faculty of Medicine, Addis Ababa University. Ethical approval was obtained from SPHMMC to collect archived FFPE tissue samples from enrolled patients. The study was also approved by the National Research Ethics Review Committee at the Ethiopian ministry of Science and Technology. Written informed consent was obtained from every patient and all tissue samples were fully anonymized before accessed. Medical records of patients who undergone surgery between October 2012 and December 20l5 were accessed to collect sociodemographic and pathological data. These cohorts were also used for our previous published work aimed at assessment of the frequency and distribution of molecular subtypes of breast cancer in Ethiopian women [26].

This study was a cross sectional retrospective in design and recruited participants who visited the Oncology Centre at the Tikur Anbessa Specialized Hospital (TASH) which is referral hospital found in Addis Ababa, Ethiopia. Patients were enrolled based on availability of FFPE tissue either at TASH or St. Paul's Hospital Millennium Medical College (SPHMMC) which is also a public referral hospital in the capital where some of our participants had their pathology tests done. Patient medical records were accessed by the investigators to collect variables which includes age, tumor type, grade and stage of disease. Initially, a total of 189 patients were included in the study but 66 cases were excluded because of issues related to availability or quality of FFPE tissue samples. The final number of cases consisted of 123 patients and all archived FFPE blocks were sectioned, H&E stained and examined by a pathologist at Orebro Hospital in Sweden for locating tumors in the block to be used for constructing tissue microarray (TMA). The total number of cases, after TMA construction, to proceed with the molecular analysis were 114.

Before TMA construction, digital images for constructing TMA were taken from slides scanned with a Pannoramic 250 digital scanner (3D HISTECH Ltd., Budapest, Hungary) and representative areas selected from images using the software program 'Case viewer' (3D HISTECH Ltd., Budapest, Hungary). TMA grand master automated system (3DHISTECH Ltd., Budapest, Hungary) were used to construct the TMA. The size of TMA biopsy punches was 0.6-millimeter and it was prepared in triplicate from corresponding area marked by a pathologist. The punches were then taken from donor paraffin blocks and merged into TMA recipient paraffin blocks.

TMA slides were stained with monoclonal antibodies specified in "Table 1" following standard protocols in automated system using the Dako Autostainer Link. Briefly, following deparaffinization and rehydration, heat induced epitope retrieval was performed with FLEX TRS High pH Retrieval buffer for 20 minutes. After peroxidase blocking, the specific monoclonal antibodies were applied at room temperature for 20 minutes. Detection was made using the FLEX + Rabbit EnVision System. DAB chromogen was then applied for 10 minutes. Finally, Slides were counterstained with Mayers hematoxylin for 5 seconds and then dehydrated and coverslipped.

Amplification of HER2 was evaluated using PATH Vysion (HER-2/CEP17) FISH Probe Kit from Abbott Molecular, Des Plaines, IL. Briefly, slides for FISH testing were deparaffinized, rinsed in absolute alcohol, and air dried. The sections were then subjected to pretreatment according to the manufacturer's protocol. Slides were hybridized with a probe mix in HYBrite

**Table 1. Sources and dilutions of primary antibodies.**

| Antibody | Clone | Manufacturer | Dilution |
|---|---|---|---|
| ER | EP1 | Agilent Dako | RTU |
| PR | PgR1294 | Agilent Dako | RTU |
| Ki-67 | Mib-1 | Agilent Dako | RTU |
| Her-2 | Herceptest | Agilent Dako | RTU |
| AR | EP120 | Epitomics | 1/100 |

RTU = Ready to use

(Vysis, Des Plaines, IL) where denaturation was set at 6 min at 73˚ C and hybridization for 17 hr at 37˚ C. Images were scored using the software program 'Case viewer' (3D HISTECH Ltd., Budapest, Hungary) after slides were scanned on a Pannoramic MIDI digital scanner (3D HISTECH Ltd., Budapest, Hungary). Signals from 20 tumor cells were evaluated for scoring HER2 amplification.

The expression of ER, PR, and HER2 were evaluated and scored using a standard criterion. A cutoff value of 1% was used to define ER and PR positivity based on the ASCO/CAP 2013 guidelines [27]. HER2 was graded based on the degree of membrane staining, on a scale of 0–3 based on recommendations from Fitzgibbons et. al [28]. Grades of 0–1+ are considered negative, a grade of 2+ is equivocal, and a grade of 3+ is considered positive for HER2 labeling. FISH for HER-2 amplification was considered positive when HER2:CEP17 ratios is $\geq$ 2:0. A Ki67 cut-off value of 20% in a minimum of 500 cells was used to define a high score as described in St. Gallen international panel of experts' recommendation [29]. A cutoff value of 1% nuclear staining was used to define AR positivity irrespective of intensity as previously described by Asano et.al. [30, 31].

Based on the IHC results and confirmation of HER2 amplification by FISH analysis., the tumors were classified into the following four molecular sub-types according to the St. Gallen international expert's consensus 2013 [29]: luminal A (ER and/or PR-positive, HER2-negative and Ki67<20%), luminal B (ER and/or PR-positive, HER2-positive OR ER and/or PR-positive, HER2-negative and Ki67$\geq$20%), HER2-enriched (ER and PR-negative, HER2 positive) and triple-negative (ER-negative, PR-negative and HER2-negative).

## Statistical analysis

Statistical analysis was done using SPSS for windows version 21. Continuous data are reported as mean ± SD or Number (proportions). Skew distributions are reported as the median value with minimum and maximum. All P values are two tailed and P value < 0.05 was considered statistically significant. The Mann-Whitney U test was used to compare median age. Chi-square test and ANOVA were used to determine correlations.

## Results and discussion

There were 114 participants with samples available in acceptable quality for pathological and molecular test in this study. Mean age at diagnosis of the participants was 43 years (SD 14) and median age was 40 (range 22–75). Most of the study participants (40%) were < 40 years old. In this study, 31% of the participants were $\geq$ 50 years and only 19% were 40–49 years old. "Table 2" shows basic pathological and molecular characteristics of the study subjects.

A total of 112 cases had complete data concerning AR immunostaining. These cases were classified as positive and negative as previously done by Asano et al., 2017 [30]. A

**Table 2. Baseline clinicopathological and molecular characteristics of the study participants (n = 114).**

| Variables | N (%) |
|---|---|
| **Histological Grade** | |
| I | 7 (6) |
| II | 32 (28) |
| III | 39 (34) |
| Missing | 36 (32) |
| **Estrogen Receptor** | |
| Positive | 74 (65) |
| Negative | 40 (35) |
| **Progesterone Receptor** | |
| Positive | 49 (43) |
| Negative | 64 (56) |
| Missing | 1 (1) |
| **HER2** | |
| Positive | 26 (23) |
| Negative | 87 (76) |
| Missing | 1 (1) |
| AR | |
| Positive | 91 (80) |
| Negative | 21 (18) |
| Missing | 2 (2) |
| **Histological Type** | |
| Infiltrating Ductal | 67 (59) |
| Lobular | 6 (5) |
| Others/Not classified | 25 (22) |
| Missing | 16 (14) |
| **Stage** | |
| I | 19 (17) |
| II | 37 (33) |
| III | 36 (31) |
| IV | 4 (3) |
| Missing | 18 (16) |

representative image of the immunostaining is presented in "Fig 1". Out of 112 participants, 91 (81%) were positive for AR expression. The remaining 21 participants (19%) were negative for AR expression. "Tables 3 and 4" shows the distribution of clinicopathological and molecular parameters in AR expressing and non-expressing tumors. "Tables 3 and 4" also explores association of AR expression with variable clinical and molecular parameters among AR+ and AR- tumors. There was no statistically significant difference in the distribution of the clinicopathological parameters between AR expressing and non-expressing tumors.

In this study, AR was expressed in 81% of breast tumors which is higher than the expression rates of both ER and PR. This result is comparable to previous studies conducted in other parts of the world mainly in the western countries (70–90%) [8, 14, 17, 21–23]. However, our result is different than the two African studies conducted in Ghana and Tanzania which was 44% and 66% respectively [24, 25]. The reason for the variation between our study (overall AR-positivity of 81%) and the Tanzanian study (overall AR-positivity of 66%) could be due to true biological difference in tumor characteristics in the two countries or methodological difference

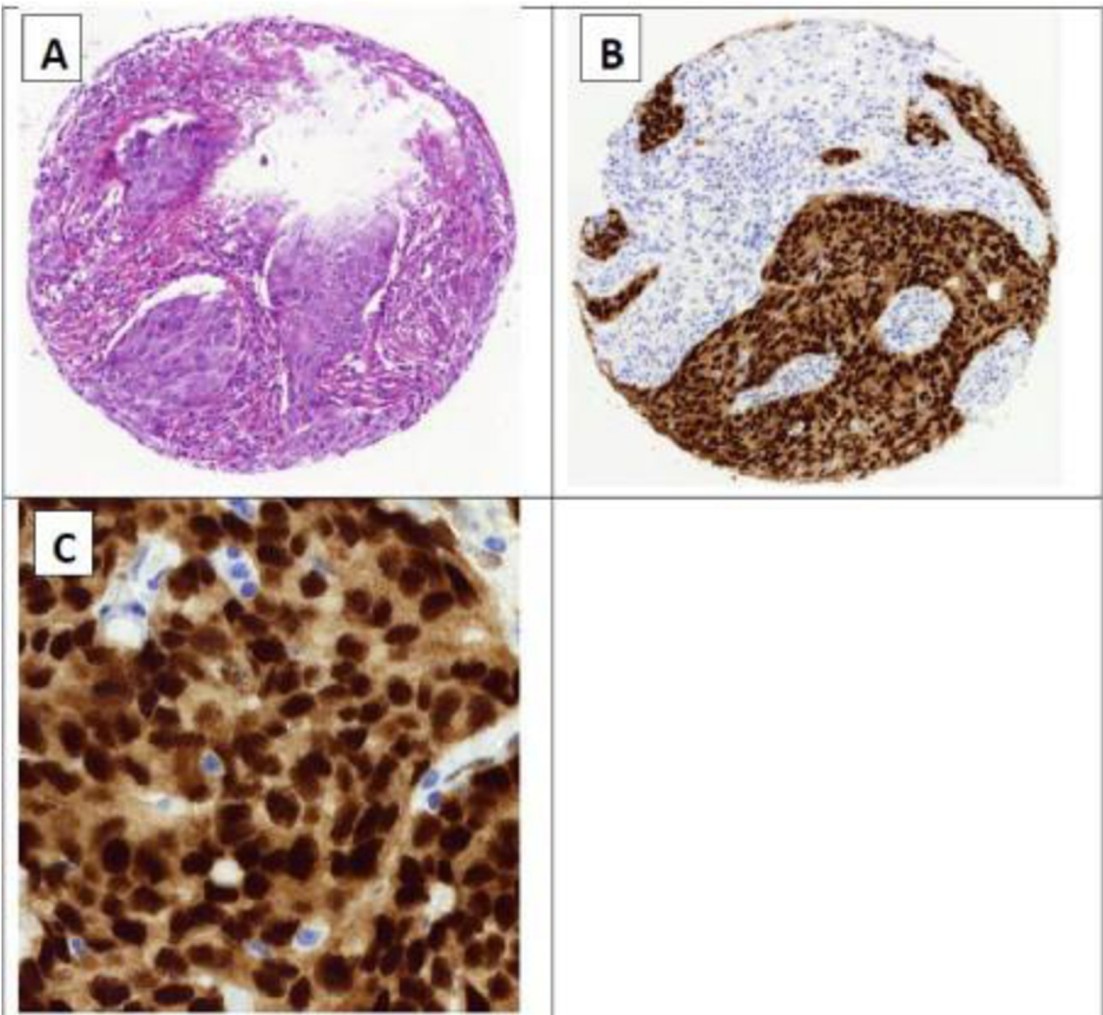

**Fig 1. Representative images of AR positive immunohistochemistry in breast tumor tissue array. A**, H&E staining (10X); **B**, immunostaining (10X); **C**, immunostaining of a specific area (40X).

including variation in antibody clones used and sample size The variation in AR-positivity between our study and the Ghanaian study (overall AR-positivity 44%) could be also be both technical and biological differences because the Ghanaian study used >10% cut-off value for AR-positivity and it is obvious that the frequency of AR-positivity would have been much higher if the now commonly used >1% cut-off value which was applied in our study has been applied in their analysis.

Our study showed 93% of ER positive breast cancers cases also express AR. This is in line with previous studies among Caucasians (84–95%) but we couldn't compare our result with the Tanzania study because the investigators didn't present comparison by ER status [8, 14, 17, 21–23]. Similarly, comparison by ER status between our study and the Ghanaian study were not possible because the researchers didn't present their data separately for ER rather were reported for ER/PR as an aggregate. In our study, there was a statistically significant variation (P<0.05) in the proportion of AR positivity between ER-positive (93%) and ER-negative (60%) tumors. This result is in line with previous observations in the western studies and reveals majority of ER positive tumors in our cohort co-express AR which according to

**Table 3. Association between AR expression and clinicopathological parameters among the study participants.**

| Clinico-pathological parameters | AR Negative | AR Positive | Total | p-value |
|---|---|---|---|---|
| **Median age at Diagnosis(min-max)** | 44 (29–70) | 40 (22–75) | | 0.218 |
| **Tumor Grade, n (%)** | | | | |
| I | 1 (6) | 6 (10) | 7 (9) | |
| II | 8 (47) | 23 (39) | 31 (39) | 0.774 |
| III | 8 (47) | 30(51) | 38 (48) | |
| **Stage, n (%)** | | | | |
| I | 3 (16) | 16 (21) | 19 (20) | |
| II | 5 (26) | 30 (41) | 35 (38) | 0.507 |
| III | 10 (53) | 26 (34) | 36 (38) | |
| IV | 1 (3) | 3 (4) | 4 (4) | |
| **Histological Type, n (%)** | | | | |
| Infiltrating ductal | 17 (85) | 49 (65) | 66 (69) | 0.169 |
| Lobular | 0 (0) | 6 (8) | 6 (6) | |
| Others/Unknown | 3 (15) | 21 (27) | 24 (25) | |

accumulating evidences has beneficiary role [8, 14, 17, 21–23]. We were unable to do comparison between our study and the two African studies for AR expression with ER status because of absence of such information in those studies. There was a statistically significant variation (P<0.05) in the proportion of AR positive cases between PR-positive (98%) and PR-negative (70%) tumors as well. This is also in line with previous observations [32]. There was no statistically significant variation in the expression of AR between HER2-positive and HER2-negative tumors in this study (P = 0.145). About 83% of HER2+/ER- tumors were positive for AR in this study which is higher than previous studies among Caucasians (50–63%) [33, 34]. The reason for this difference could be due to small number of HER2+/ER- tumors (only 12 cases) in our study.

The expression of AR among TNBC in our study was 48% which is comparable with both the Tanzanian study (54%) and reports among Caucasians (10–53%) [30, 35] but is

**Table 4. Association between AR and molecular parameters among the study participants.**

| Molecular parameters | AR Negative | AR Positive | Total | p-value |
|---|---|---|---|---|
| **ER, n (%)** | | | | |
| Negative | 15 (75) | 23 (25) | 38 (34) | 0.000* |
| Positive | 5 (25) | 68 (75) | 73 (66) | |
| **PR, n (%)** | | | | |
| Negative | 19 (95) | 44 (48) | 63 (53) | |
| Positive | 1 (5) | 47 (52) | 48 (43) | 0.000* |
| **HER2, n (%)** | | | | |
| Negative | 17 (89) | 67 (74) | 84 (76) | |
| Positive | 2 (11) | 24 (26) | 26 (24) | 0.145 |
| **Molecular Subtype, n (%)** | | | | |
| Luminal A | 4 (21) | 41 (45) | 45 (41) | |
| Luminal B | 1 (5) | 28 (31) | 29 (26) | |
| HER2-enriched | 1 (5) | 10 (11) | 11 (10) | 0.000* |
| TNBC/basal-like | 13 (68) | 12 (13) | 25 (23) | |

*P≤0.05

significantly different than the proportion among Nigerian TNBC patients (8.3%) reported by Bhattarai et. al. in their recent global AR study [36]. The difference in AR expression among TNBC patients between our study and Nigerians in the Bhattarai et. al. study could be population variation in AR expression among the two countries. The expression of AR in our study among the other molecular subtypes was significantly different ($P<0.05$) which is also in line with previous studies in the western countries [8, 32, 37]. No statistically significant correlation was found between the clinicopathological parameters and AR expression in this study.

## Conclusion

Androgen receptor (AR) is expressed in a significant number of most types of breast cancers and is more frequently expressed than ER and PR. Our study shows AR expression is significantly high among ER+ breast cancer patient. Similarly, AR is expressed in a significant number of triple-negative breast cancers. These indicates that breast cancers patients from Ethiopia may have favorable prognosis and could also benefit from progresses in AR targeted treatments under development. Since AR expression has important consequences on the prognosis and treatment of breast cancer, further studies with an increased number of samples is necessary to confirm our reports.

## Supporting information

**S1 Data.**
(XLSX)

## Acknowledgments

We acknowledge TASH and SPHMMC pathology laboratories for facilitating access to archived FFPE tissues. We acknowledge AHRI for facilitating the study by continuously mentoring and proving administrative support to the PI. We acknowledge University of Michigan Center for International Reproductive Health (CIRHT) project for arranging relevant training for the PI in USA. We would like also to acknowledge Orebro University in Sweden for allowing the PI to have Laboratory access to conduct the analysis. We acknowledge Endegena Abebe and Sisay Addisu (PhD students at AAU) for supporting collection of some of the samples. We also acknowledge Elin Embretsen-Varro and Anna Gotrilin-Eremo in Orebro University Hospital for technical support. We also like to acknowledge Dr Maheteme Bekele and Dr Aisha Jibril at SPHMMC for facilitating access to some of the tissue blocks used in this study from SPHMMC pathology laboratory.

## Author Contributions

**Conceptualization:** Endale Hadgu, Daniel Seifu, Wondemagegnhu Tigneh, Yonas Bokretsion, Abebe Bekele, Markos Abebe, Christina Karlsson, Mats G. Karlsson.

**Data curation:** Endale Hadgu, Daniel Seifu, Yonas Bokretsion, Abebe Bekele, Markos Abebe, Thomas Sollie, Christina Karlsson, Mats G. Karlsson.

**Formal analysis:** Endale Hadgu, Daniel Seifu, Christina Karlsson, Mats G. Karlsson.

**Funding acquisition:** Daniel Seifu, Markos Abebe.

**Investigation:** Endale Hadgu, Daniel Seifu, Wondemagegnhu Tigneh, Thomas Sollie, Christina Karlsson, Mats G. Karlsson.

**Methodology:** Endale Hadgu, Daniel Seifu, Yonas Bokretsion, Abebe Bekele, Christina Karlsson, Mats G. Karlsson.

**Project administration:** Endale Hadgu, Daniel Seifu, Markos Abebe, Christina Karlsson, Mats G. Karlsson.

**Resources:** Daniel Seifu, Markos Abebe, Mats G. Karlsson.

**Software:** Christina Karlsson.

**Supervision:** Daniel Seifu, Wondemagegnhu Tigneh, Yonas Bokretsion, Abebe Bekele, Markos Abebe, Christina Karlsson, Mats G. Karlsson.

**Validation:** Endale Hadgu, Wondemagegnhu Tigneh, Thomas Sollie.

**Visualization:** Endale Hadgu, Thomas Sollie, Christina Karlsson, Mats G. Karlsson.

**Writing – original draft:** Endale Hadgu.

**Writing – review & editing:** Endale Hadgu, Daniel Seifu, Wondemagegnhu Tigneh, Yonas Bokretsion, Abebe Bekele, Markos Abebe, Thomas Sollie, Christina Karlsson, Mats G. Karlsson.

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
