## [Decision Letter · Decision Letter 0]

18 Dec 2019

PONE-D-19-29283

Distribution and characteristics of androgen receptor (AR) in breast cancer among women in Addis Ababa, Ethiopia: A cross sectional study

PLOS ONE

Dear Dr. Gebregzabher,

Thank you for submitting your manuscript to PLOS ONE. After careful consideration, we feel that it has merit but does not fully meet PLOS ONE’s publication criteria as it currently stands. Therefore, we invite you to submit a revised version of the manuscript that addresses the points raised during the review process.

We would appreciate receiving your revised manuscript by Feb 01 2020 11:59PM. To enhance the reproducibility of your results, we recommend that if applicable you deposit your laboratory protocols in protocols.io, where a protocol can be assigned its own identifier (DOI) such that it can be cited independently in the future. For instructions see: http://journals.plos.org/plosone/s/submission-guidelines#loc-laboratory-protocols

We look forward to receiving your revised manuscript.

Kind regards,

Tomi F. Akinyemiju, Ph.D

Academic Editor

PLOS ONE

Journal Requirements:

2. We noticed minor instances of text overlap with the following previous publication(s), which need to be addressed:

https://bmcwomenshealth.biomedcentral.com/articles/10.1186/s12905-018-0531-2

 The text that needs to be addressed involves the Abstract and the Methods section.

 In your revision please ensure you cite all your sources (including your own works), and quote or rephrase any duplicated text. Further consideration is dependent on these concerns being addressed.

3. Please provide additional details regarding participant consent. In the Methods section, please ensure that you have specified (1) whether consent was informed and (2) what type you obtained (for instance, written or verbal). If your study included minors, state whether you obtained consent from parents or guardians. If the need for consent was waived by the ethics committee, please include this information.

4. In the ethics statement in the manuscript and in the online submission form, please provide additional information about the patient records used in your retrospective study, including: a) whether all tissue samples were fully anonymized before you accessed them; b) the date range (month and year) during which patients' medical records were accessed; c) the source of the medical records analyzed in this work (e.g. hospital, institution or medical center name). If patients provided informed written consent to have data from their medical records used in research, please include this information.

5. Please amend either the title on the online submission form (via Edit Submission) or the title in the manuscript so that they are identical.

6. Please upload a copy of Figure 1, to which you refer in your text on page 9. If the figure is no longer to be included as part of the submission please remove all reference to it within the text.

Reviewers' comments:

Reviewer's Responses to Questions

**Comments to the Author**

1. Is the manuscript technically sound, and do the data support the conclusions?

Reviewer #1: Partly

Reviewer #2: Partly

2. Has the statistical analysis been performed appropriately and rigorously? 

Reviewer #1: No

Reviewer #2: Yes

3. Have the authors made all data underlying the findings in their manuscript fully available?

Reviewer #1: Yes

Reviewer #2: Yes

4. Is the manuscript presented in an intelligible fashion and written in standard English?

Reviewer #1: No

Reviewer #2: No

5. Review Comments to the Author

Reviewer #1: The manuscript by Gebregzabher et al., explores the overall androgen receptor (AR) expression in breast cancer cohort of African ancestry. The manuscript has done a great job explaining expression of AR in different breast cancer subtypes in African population. However, the manuscript lacks novelty and has some major weaknesses and my comments are as below:

1. While most clinical trial report the use of specific antibody (AR 441 from DAKO) to address the antibody related controversy over variable AR expression, the authors’ do not provide any rationale for using a different antibody for staining AR. Moreover, looking at micrographs showing AR staining in Figure 1, it seems that there is a huge amount of background/non-specific staining and scoring these type of staining may have a false positive results. The authors might have to provide micrographs for positive and negative controls used.

2. Multiple factors like sample procuring method, antibody used to stain and method of staining and scoring (different cut off values) dictate the discrepancy in the results of expression of AR in TNBC. The paper do not address any of these issues.

3. There is a recent paper from Bhattarai et.al on prognostic role of AR in global TNBC cohort where the authors talk about AR expression in various cohorts including African cohort. It would be worth including the paper in the discussion.

4. Since Breast cancer patients of African ancestry have aggressive TNBC phenotype, it would be worth looking at the survival trend between AR+ and AR- TNBCs.

5. In addition, the manuscript has instances of awkward English and stylistic concerns so it would be great if the authors pay a little attention on spelling and sentence constructions.

Reviewer #2: The authors have conducted a study on the distribution of AR in BRCa among women in Ethiopa. The reasons for undertaing this study were to both continue previous work and to demonstrate potential molecular differences in BRCa in women in Ethiopa as compared with women in other African countries and Caucasian women. The study hopes to show that AR expression may have important consequences on the prognosis and treatment of breast cancer. The novelty of this study lies mostly in the fact that the study is being conducted in the Ethiopian population.

The current literature supports the idea that distribution of AR may differ by subtype and has also shown a potential for AR to be associated with positive prognosis in ER positive breast tumors. This work also shows a difference in distribution of the AR among BCa molecular subtypes, particularly associated with ER and PR positive tumors. It also shows that 80% of tumors in the study expressed AR as defined by >1% positivity.

Table 3 shows that there are no significant differences in clinico-pathological parameters between tumors that express AR and those that do not. If this is the case, what does this say about the role that AR plays in tumor behavior in Ethiopian BCa? How will therapies that target AR make a difference? It seems that this data supports that AR is a marker that is present in the majority of Ethiopian BCa, but it does not say anything about its prognostic usefulness.

Given the number of samples and the percentage of samples that are AR positive, the authors have already agreed that the study would be stronger with the increase in number of samples. It is possible that the low number of AR negative samples are masked in the measure of differences in clinico-pathological parameters.

Therefore, while I agree that the study shows, as others have shown, that AR expression is high in ER positive BCa, I do not agree that THIS study shows that Ethiopian BCa could benefit from AR targeted treatments. It would be helpful if the study could show survival or treatment data that would allow a clearer picture of the prognostic value of AR in this population. Increasing the numbers of samples may indeed show significant differences in clinical parameters.

The submitted manuscript needs significant editing regarding grammar and clarity. The first part of the discussion is almost a word-for-word repeat of the introduction instead of a summary of the study and overall findings. Also, there is no need to include a statement saying that you aren't going to re-hash the methods portion of the previous study. Either briefly describe the methods from the previous study with explanation of AR specific portions of these methods and reference to your previous study, or re-write the methods. It seems that you have re-written the methods anyway.

Please do not use the word "about" when you are referring to the number of samples. Give the absolute number and describe in detail what it means for a sample to be "evaluable."

6. PLOS authors have the option to publish the peer review history of their article (what does this mean?). If published, this will include your full peer review and any attached files.

Reviewer #1: No

Reviewer #2: No

---

## [Author Response · Author response to Decision Letter 0]

11 Jan 2020

Reviewer 1 

We thank the reviewer for the valuable comments! 

Please find our responses: 

1. While most clinical trial report the use of specific antibody (AR 441 from DAKO) to address the antibody related controversy over variable AR expression, the authors’ do not provide any rationale for using a different antibody for staining AR. Moreover, looking at micrographs showing AR staining in Figure 1, it seems that there is a huge amount of background/non-specific staining and scoring these type of staining may have a false positive results. The authors might have to provide micrographs for positive and negative controls used.

>>> We don’t have any special reason to use the Anti androgen receptor antibody clone EP 120 from Epitomics. It was the clone available in the laboratory at the time. We have supplied the micrograph for the positive and negative controls used in the study.

2. Multiple factors like sample procuring method, antibody used to stain and method of staining and scoring (different cut off values) dictate the discrepancy in the results of expression of AR in TNBC. The paper do not address any of these issues.

>>> The comment is correct, and we have addressed the issue under our discussion part in paragraph 4 page 12.

3. There is a recent paper from Bhattarai et.al on prognostic role of AR in global TNBC cohort where the authors talk about AR expression in various cohorts including African cohort. It would be worth including the paper in the discussion.

>>>We have included the paper in our discussion.

4. Since Breast cancer patients of African ancestry have aggressive TNBC phenotype, it would be worth looking at the survival trend between AR+ and AR- TNBCs.

>>>This is a very good recommendation and would consider comparing survival in other study.

5. In addition, the manuscript has instances of awkward English and stylistic concerns so it would be great if the authors pay a little attention on spelling and sentence constructions.

>>>We think this is addressed in the revision

Reviewer 2 

We thank the reviewer for valuable comments! 

Please find our responses: 

1. Table 3 shows that there are no significant differences in clinico-pathological parameters between tumors that express AR and those that do not. If this is the case, what does this say about the role that AR plays in tumor behavior in Ethiopian BCa? How will therapies that target AR make a difference? It seems that this data supports that AR is a marker that is present in the majority of Ethiopian BCa, but it does not say anything about its prognostic usefulness. Given the number of samples and the percentage of samples that are AR positive, the authors have already agreed that the study would be stronger with the increase in number of samples. It is possible that the low number of AR negative samples are masked in the measure of differences in clinico-pathological parameters.

Therefore, while I agree that the study shows, as others have shown, that AR expression is high in ER positive BCa, I do not agree that THIS study shows that Ethiopian BCa could benefit from AR targeted treatments. It would be helpful if the study could show survival or treatment data that would allow a clearer picture of the prognostic value of AR in this population. Increasing the numbers of samples may indeed show significant differences in clinical parameters.

>>> This is correct. Previous studies assessing the role of AR in breast cancer showed its positive prognostic role in hormone receptor positive BCa and its therapeutic significance in TNBC and is presented in our paper. The lack of association between AR expression and clinicaopathological parameters in our study is highly likely due to small sample size and is stated in the discussion part of the paper. We agree that survival or treatment data would give a clear picture and hope future studies to address these issues.

2. The submitted manuscript needs significant editing regarding grammar and clarity. The first part of the discussion is almost a word-for-word repeat of the introduction instead of a summary of the study and overall findings. Also, there is no need to include a statement saying that you aren't going to re-hash the methods portion of the previous study. Either briefly describe the methods from the previous study with explanation of AR specific portions of these methods and reference to your previous study, or re-write the methods. It seems that you have re-written the methods anyway.

>>> We agree with your concern and have edited the paper in the resubmission.

3. Please do not use the word "about" when you are referring to the number of samples. Give the absolute number and describe in detail what it means for a sample to be "evaluable."

>>> We agree with your concern and removed the word “about” and have added more information to explain what evaluable is.

---

## [Decision Letter · Decision Letter 1]

11 Mar 2020

PONE-D-19-29283R1

Distribution and characteristics of androgen receptor (AR) in breast cancer among women in Addis Ababa, Ethiopia: A cross sectional study

PLOS ONE

Dear Dr. Gebregzabher, 

Thank you for submitting your manuscript to PLOS ONE. After careful consideration, we feel that it has merit but does not fully meet PLOS ONE’s publication criteria as it currently stands. Therefore, we invite you to submit a revised version of the manuscript that addresses the points raised during the review process.

Please conduct a thorough grammar check and copy-editing of the entire manuscript prior to resubmission

We would appreciate receiving your revised manuscript by Apr 25 2020 11:59PM. To enhance the reproducibility of your results, we recommend that if applicable you deposit your laboratory protocols in protocols.io, where a protocol can be assigned its own identifier (DOI) such that it can be cited independently in the future. For instructions see: http://journals.plos.org/plosone/s/submission-guidelines#loc-laboratory-protocols

We look forward to receiving your revised manuscript.

Kind regards,

Tomi F. Akinyemiju, Ph.D

Academic Editor

PLOS ONE

Reviewers' comments:

Reviewer's Responses to Questions

**Comments to the Author**

1. If the authors have adequately addressed your comments raised in a previous round of review and you feel that this manuscript is now acceptable for publication, you may indicate that here to bypass the “Comments to the Author” section, enter your conflict of interest statement in the “Confidential to Editor” section, and submit your "Accept" recommendation.

Reviewer #1: (No Response)

Reviewer #2: All comments have been addressed

Reviewer #3: (No Response)

2. Is the manuscript technically sound, and do the data support the conclusions?

Reviewer #1: Partly

Reviewer #2: (No Response)

Reviewer #3: Partly

3. Has the statistical analysis been performed appropriately and rigorously? 

Reviewer #1: No

Reviewer #2: (No Response)

Reviewer #3: Yes

4. Have the authors made all data underlying the findings in their manuscript fully available?

Reviewer #1: Yes

Reviewer #2: (No Response)

Reviewer #3: No

5. Is the manuscript presented in an intelligible fashion and written in standard English?

Reviewer #1: No

Reviewer #2: (No Response)

Reviewer #3: No

6. Review Comments to the Author

Reviewer #1: The manuscript still has instances of awkward english and foremost this is not a novel study as there are studies assessing the expression of AR in African women. The sample size is very low and unless the survival analysis is performed we cannot mark the importance of this study. It would be impactful if the authors increase the sample size and see if there is any association of AR expression with clinicopathological parameters and perform the survival analysis to evaluate its prognostic significance.

Reviewer #2: (No Response)

Reviewer #3: The present study assesses the prevalence of AR-positive breast cancer cases among 112 newly diagnosed patients at a hospital in Addis Ababa, Ethiopia. The authors found that AR-positive expression was highly prevalent in this sample and that AR expression differed by subtype. The present draft is a revision from a previous submission. My review takes into consideration the previous reviewers' comments and the authors' corresponding revision.

Skewed continuous variables (age) should be tested for differences using a non-parametric test, such as the Wilcoxon rank-sum test.

The Materials and Methods section repeats text verbatim from earlier published work. Paraphrasing is needed. Additionally, the citation to the earlier work can be referenced once at the beginning of the Methods section and noted that this paper briefly describes those same methods.

The largest issue to address is in the Discussion when relating the present study to the literature. While differences may be due to true differences across sub-Saharan African populations, there are other possible explanations that should be addressed. This includes the use of an antibody in this study that may result in an overestimate of AR prevalence compared to other work with AR 441. Could this be why the prevalence of AR-positive BCa is higher in this study compared to others? A second factor, as the authors point out, is the cutoff score of 1% versus 10%. Is it possible with the present data to provide AR-positive prevalence using the 10% cutoff to allow for a direct comparison? Would how the participants are enrolled or the samples collected impact findings across the studies?

Continued copy-editing for grammar and style is needed.

7. PLOS authors have the option to publish the peer review history of their article (what does this mean?). If published, this will include your full peer review and any attached files.

Reviewer #1: No

Reviewer #2: No

Reviewer #3: No

---

## [Author Response · Author response to Decision Letter 1]

18 Mar 2020

Reviewer 1 

We thank the reviewer for the valuable comments! 

Please find our responses: 

1. The manuscript still has instances of awkward english and foremost this is not a novel study as there are studies assessing the expression of AR in African women. The sample size is very low and unless the survival analysis is performed we cannot mark the importance of this study. It would be impactful if the authors increase the sample size and see if there is any association of AR expression with clinicopathological parameters and perform the survival analysis to evaluate its prognostic significance.

>>> We have made an extensive language editing in this revision and hope to answer your question.

>>> Concerning sample size, we acknowledge an increased sample size would provide a better insight into the study. We clearly indicated this as a one of the limitations and hope would be addressed by future studies.

>>>We agree survival study could reveal a better information, but we don’t have data to do survival analysis.

Reviewer 3 

We thank the reviewer for valuable comments! 

Please find our responses: 

1. Skewed continuous variables (age) should be tested for differences using a non-parametric test, such as the Wilcoxon rank-sum test.

>>> This is correct and addressed in the revision.

2. The Materials and Methods section repeats text verbatim from earlier published work. Paraphrasing is needed. Additionally, the citation to the earlier work can be referenced once at the beginning of the Methods section and noted that this paper briefly describes those same methods.

>>> We agree with your concern and have edited the paper in the revision.

3. The largest issue to address is in the Discussion when relating the present study to the literature. While differences may be due to true differences across sub-Saharan African populations, there are other possible explanations that should be addressed. This includes the use of an antibody in this study that may result in an overestimate of AR prevalence compared to other work with AR 441. Could this be why the prevalence of AR-positive BCa is higher in this study compared to others? A second factor, as the authors point out, is the cutoff score of 1% versus 10%. Is it possible with the present data to provide AR-positive prevalence using the 10% cutoff to allow for a direct comparison? Would how the participants are enrolled or the samples collected impact findings across the studies?

>>> We agree with you concerning the antibody clone and have included your indication in the revision.

>>> AR is currently interpreted at a cutoff of 1% and it is our hope that more studies would be conducted on AR in this continent and a future comparative study would possibly come up with a better insight.

With kind regards!

Endale Hadgu

---

## [Editor Report · Decision Letter 2]

31 Mar 2020

PONE-D-19-29283R2

Distribution and characteristics of androgen receptor (AR) in breast cancer among women in Addis Ababa, Ethiopia: A cross sectional study

PLOS ONE

Dear Dr. Gebregzabhar,

Thank you for submitting your manuscript to PLOS ONE. After careful consideration, we feel that it has merit but does not fully meet PLOS ONE’s publication criteria as it currently stands. Therefore, we invite you to submit a revised version of the manuscript that addresses the points raised during the review process.

We would appreciate receiving your revised manuscript by June 27, 2020. To enhance the reproducibility of your results, we recommend that if applicable you deposit your laboratory protocols in protocols.io, where a protocol can be assigned its own identifier (DOI) such that it can be cited independently in the future. For instructions see: http://journals.plos.org/plosone/s/submission-guidelines#loc-laboratory-protocols

We look forward to receiving your revised manuscript.

Kind regards,

Tomi F. Akinyemiju, Ph.D

Academic Editor

PLOS ONE

Additional Editor Comments (if provided):

Whole sentences and sections from the BMC Women's Health paper are still presented in the current manuscript verbatim. Even while referencing the previous manuscript and stating the overlap, there still needs to be extensive paraphrasing and re-wording so that the text is not exactly the same across the two publications. If this is done, there is no need to include this sentence:

'Therefore, the description of the methods presented here though rephrased may overlap with our previous paper with the exception of AR specific information.'

Further grammatical editing is still highly suggested to improve the quality of the writing/language

---

## [Author Response · Author response to Decision Letter 2]

1 Apr 2020

Dear Editors, 

We would like to thank you again for giving us the opportunity to re-submit our manuscript to PLOS ONE. Thank you very much for reviewing the paper and giving valuable feedback. We have addressed the academic editor comments in the resubmitted draft, with changes highlighted. We would like to thank the academic editor for their constructive ideas which have helped improve comprehension for the audience. 

We provide here a letter with a point-by-point response to the concerns: 

Response to academic editor:

1. Whole sentences and sections from the BMC Women's Health paper are still presented in the current manuscript verbatim. Even while referencing the previous manuscript and stating the overlap, there still needs to be extensive paraphrasing and re-wording so that the text is not exactly the same across the two publications. If this is done, there is no need to include this sentence:

'Therefore, the description of the methods presented here though rephrased may overlap with our previous paper with the exception of AR specific information.'

Further grammatical editing is still highly suggested to improve the quality of the writing/language

>>> We have made the language editing as per your advice in this revision and hope to answer your question.

With kind regards!

Endale Hadgu

---

## [Editor Report · Decision Letter 3]

17 Apr 2020

Distribution and characteristics of androgen receptor (AR) in breast cancer among women in Addis Ababa, Ethiopia: A cross sectional study

PONE-D-19-29283R3

Dear Dr. Gebregzabher,

We are pleased to inform you that your manuscript has been judged scientifically suitable for publication and will be formally accepted for publication once it complies with all outstanding technical requirements.

With kind regards,

Tomi F. Akinyemiju, Ph.D

Academic Editor

PLOS ONE
---

## [Editor Report · Acceptance letter]

22 Apr 2020

PONE-D-19-29283R3 

Distribution and characteristics of androgen receptor (AR) in breast cancer among women in Addis Ababa, Ethiopia: A cross sectional study 

Dear Dr. Gebregzabher:

I am pleased to inform you that your manuscript has been deemed suitable for publication in PLOS ONE. Congratulations! Your manuscript is now with our production department. 

With kind regards,

on behalf of

Dr. Tomi F. Akinyemiju 

Academic Editor

PLOS ONE